# Porcine Intestinal Apical-Out Organoid Model for Gut Function Study

**DOI:** 10.3390/ani12030372

**Published:** 2022-02-03

**Authors:** Sang-Seok Joo, Bon-Hee Gu, Yei-Ju Park, Chae-Yun Rim, Min-Ji Kim, Sang-Ho Kim, Jin-Ho Cho, Hyeun-Bum Kim, Myunghoo Kim

**Affiliations:** 1Department of Animal Science, College of Natural Resources & Live Science, Pusan National University, Miryang 50463, Korea; ssjoo7680@gmail.com (S.-S.J.); zx7777777@pusan.ac.kr (Y.-J.P.); yeoul00h@naver.com (C.-Y.R.); 2Life and Industry Convergence Research Institute, Pusan National University, Mirayng 50463, Korea; g.bonhee@gmail.com; 3Animal Nutrition and Physiology Division, National Institute of Animal Science, Rural Development Administration, Wanju 55365, Korea; mjkim00@korea.kr (M.-J.K.); kims2051@gmail.com (S.-H.K.); 4Department of Animal Science, Chungbuk National University, Cheongju 28644, Korea; jinhcho@cbnu.ac.kr; 5Department of Animal Resources and Science, Dankook University, Cheonan 31116, Korea; hbkim@dankook.ac.kr

**Keywords:** pig model, intestinal organoid, apical-out, nutrient uptake, barrier integrity

## Abstract

**Simple Summary:**

Pigs have been used in various animal model studies on the gastrointestinal tract (GIT) across both animal science and biomedical science fields. Recently, intestinal organoids have been used as a research tool for the GIT, and they have also been applied to farm animals, including pigs. However, to our knowledge, no functional studies of the porcine intestine using intestinal organoids have been conducted to date. In the present study, we developed two porcine intestinal organoid models (basal-out and apical-out organoids) and compared their characteristics. We also confirmed the possibility of conducting research related to intestinal functions, such as nutrient uptake and gut barrier function. The present study suggests that porcine intestinal organoids can be used as potential models for future GIT mechanism studies, such as host–microbe interactions, harmful ingredient tests, and nutritional research.

**Abstract:**

Pig models provide valuable research information on farm animals, veterinary, and biomedical sciences. Experimental pig gut models are used in studies on physiology, nutrition, and diseases. Intestinal organoids are powerful tools for investigating intestinal functions such as nutrient uptake and gut barrier function. However, organoids have a basal-out structure and need to grow in the extracellular matrix, which causes difficulties in research on the intestinal apical membrane. We established porcine intestinal organoids from jejunum tissues and developed basal-out and apical-out organoids using different sub-culture methods. Staining and quantitative real-time PCR showed the difference in axis change of the membrane and gene expression of epithelial cell marker genes. To consider the possibility of using apical-out organoids for intestinal function, studies involving fatty acid uptake and disruption of the epithelial barrier were undertaken. Fluorescence fatty acid was more readily absorbed in apical-out organoids than in basal-out organoids within the same time. To determine whether apical-out organoids form a functional barrier, a fluorescent dextran diffusion assay was performed. Hence, we successfully developed porcine intestinal organoid culture systems and showed that the porcine apical-out organoid model is ideal for the investigation of the intestinal environment. It can be used in future studies related to the intestine across various research fields.

## 1. Introduction

The intestine as mucosal tissue has multiple functions, such as nutrient digestion/uptake and immune responses against external environmental agents [1]. In the intestine, intestinal epithelial cells form a monolayer and renew themselves every three to five days; thus, it is one of the most actively regenerated tissues [2]. The epithelium in the GIT plays functions to ensure the well-controlled regeneration of intestinal stem cells (ISCs) by forming the crypt–villus tissue architecture [3]. For example, ISCs in the small intestine are able to differentiate into specific cell types, such as absorptive enterocytes, goblet cells, Paneth cells, and enteroendocrine cells. These intestinal epithelial cells are responsible for nutrient absorption, formation of mucus layer, production of antimicrobial peptides, and hormone secretion [4].

Animal models undoubtedly provide valuable information for both animal and veterinary sciences and biomedical sciences. Many in vivo pig studies have been performed to better understand the changes in physiology, productivity, and pathology as well as various factors such as infections, stress, and nutrition. Out of the various animal models, pigs are often used in human physiology research due to their anatomical and physiological similarities to humans [5]. GIT studies using pig models that aim to understand human digestive disorders and inflammatory diseases are amongst the most appropriately applied cases compared with other organ studies [6,7,8]. Although in vivo pig experiments have many advantages, they also have limitations including difficulties in manipulating the experiment, high-throughput study, and animal welfare issues. Therefore, there is a need to develop alternative approaches to animal experiments.

Recently, intestinal organoids have gained attention as advanced in vitro tools for understanding the gut environment. In livestock and veterinary research, intestinal organoid culture systems have recently been reported, including for pigs [9]. An intestinal organoid has a three-dimensional (3D) structure that mimics the intestinal epithelial environment by forming a crypt–villus structure in vitro. It can be used for various mechanistic studies on regulation of intestinal physiology and immunity [10]. For example, a culture system for porcine small intestinal organoids has been used for lentivirus transduction in functional research [11,12]. In addition, to investigate various physiological actions in GIT such as nutrient absorption, digestive enzymes and hormone secretion in pig organoids have been reported [13]. Although the 3D structure of an organoid shows high similarity to living systems in an in vitro culture system, it has some structural disadvantages. For example, a 3D organoid culture system has limited access to luminal agents, such as nutrients, chemical compounds, or microorganisms, for apical surface treatments of the organoids. This may create problems for studies on host–microbe interactions and nutrient metabolism. To avoid this limitation, modifying the form of intestinal organoids was attempted. In epithelial tight junction and permeability tests, porcine ileum organoids were used as a form of two-dimensional (2D) monolayer [14]. In this system, nutrient uptake (glucose absorption) and reactions to bacterial enterotoxins were examined [15]. Another example is a new organoid cultivation technique that was developed using suspension culture [16]. The model can cause the apical surface to face outward and can be utilized in several GIT studies. To the best of our knowledge, nutritional or gut barrier function studies that used porcine apical-out organoids have not been investigated.

Despite the importance of pig organoids in many fields of research, there is limited information on pig gut organoid systems. Thus, we established basal-out and apical-out organoid systems and compared their potential in the study of nutrition and gut barrier function. Our study suggests that the porcine apical-out organoid model can be utilized as a useful in vitro system to study physiological reactions in the porcine intestine.

## 2. Materials and Methods

### 2.1. Isolation of Crypts from Porcine Small Intestine

The crypts of porcine were harvested from the jejunum tissue of 5-week-old pigs (*n* = 3, body weight: 7.80 ± 0.96 kg). All piglets were healthy without diarrhea symptoms. A total of 3–4 cm of tissue was harvested and opened longitudinally. To remove luminal content, mucus, and part of villus, tissue was scraped using slide glass and washed with phosphate-buffered saline (PBS) 3–4 times. Next, 0.5 × 0.5 cm^2^ jejunum segments were cut and incubated for 30 min on ice at horizontal shaking of 100 rpm using 30 mM ethylene-diamine-tetra acetic acid (EDTA) and 1 mM DL-dithiothreitol (DTT). After incubation step, tissue fragments were transferred to cold crypt washing buffer (54.9 mM D-sorbitol and 43.4 mM sucrose in PBS) and vigorously shaken for 2 min to release intestinal crypts. Then, to remove villus and debris, supernatant was transferred to new 50 mL tube by using 100 μm cell strainer (SPL Life Sciences, Pocheon, Korea) and was centrifuged at 200× *g* for 2 min (two times repeat for removing debris and single cells). The crypt pellet was suspended with advanced Dulbecco’s modified Eagle medium/F12 (advanced DMEM/F12) (Thermo Fisher Scientific, Waltham, MA, USA) and counted for culturing organoids.

### 2.2. Culture of Porcine Small Intestinal Organoid

The porcine crypts were counted and mixed in Matrigel (Corning Inc., Corning, NY, USA) containing solution. These were seeded in a 96-well cell culture plate (SPL Life Sciences, Pocheon, Korea) to 5 crypts/μL concentration (total volume: 4 μL). To solidify complex, plate containing crypts in Matrigel was stored in incubator at 37 °C for 30 min. Additionally, porcine intestinal organoid culture medium (advanced DMEM/F12 supplemented with 1x N2 supplement (Thermo Fisher Scientific, Waltham, MA, USA), 1x B27 supplement (Thermo Fisher Scientific, Waltham, MA, USA), 1 mM N-acetylcysteine (Sigma-Aldrich, St. Louis, MO, USA), 50 ng/mL recombinant murine EGF (Thermo Fisher Scientific, Waltham, MA, USA), 100 μg/mL recombinant murine Noggin (Peprotech, NJ, USA), 10% R-spondin 1 conditioned media, 50% Wnt-3A conditioned media, 2 mM GlutaMAX™ Supplement (Thermo Fisher Scientific, Waltham, MA, USA), 10 mM HEPES (Thermo Fisher Scientific, Waltham, MA, USA), 100 μg/mL Primocin™ (InvivoGen, San Diego, CA, USA), 10 mM nicotinamide (Sigma-Aldrich, St. Louis, MO, USA), 10 μM Y-27632 (Selleckhem, Houston, TX, USA), 10 μM SB 202190 (Sigma-Aldrich, St. Louis, MO, USA), 0.5 μM A 83-01 (Sigma-Aldrich, St. Louis, MO, USA), and 2.5 μM CHIR99021 (Sigma-Aldrich, St. Louis, MO, USA)) was added. After first 2 days, organoids were cultured in fresh media without Y-27632, and the organoid media were replaced every 2 days.

Two critical factors for porcine small intestinal organoid culture, R-spondin 1 and Wnt3a, were utilized by using cell culture conditioned media. The conditioned media were prepared with some modifications by the method described previously [17]. To obtain conditioned media, R-spondin 1-expressing HEK 293T cell line was kindly provided from Ph.D. Lee [18]. The L Wnt-3A cell line (CRL-2647™) was purchased from American Type Culture Collection (ATCC, Manassas, VA, USA). Two types of cells were cultured in DMEM containing 10% FBS, 100 U/mL penicillin, and 100 μg/mL streptomycin with selective antibiotics (0.2 mg/mL Hygromycin B for HEK 293T cells and 0.4 mg/mL G-418 for L Wnt-3A cells). Then, advanced DMEM/F12 (containing 10% FBS, 100 U/mL penicillin and 100 μg/mL streptomycin) were used for conditioned media preparation. To gain high concentration of conditioned media, R-spondin 1 and L Wnt-3A cells were initially seeded in 5 × 10^5^ cell to 100 mm cell culture dish and cultured for 7 days. In brief, cells were grown for 4 days (approximately to 8–90% confluency) and taken off media (first batch). Then, they were replaced by fresh culture media and cultured for another 3 days and taken off media (second batch). Next, first and second batch were mixed and filtered at 1:1 ratio and stored at −20 °C until use.

### 2.3. Sub-Culture Technique and Development of Apical-Out Porcine Organoid

To passage organoids and develop apical-out organoids, Matrigel was dissociated with cell recovery solution (Corning Inc., Corning, NY, USA) at 4 °C for 30 min by using an orbital shaker (Daihan Scientific, Wonju, Korea) with gentle, slow shaking (60 rpm). Then, collected supernatant containing organoids was centrifuged at 100× *g* for 2 min and resuspended in advanced DMEM/F12 and passaged by mechanical disruption with pipetting. For basal-out organoid experiment, organoid fragments were replaced in new Matrigel and cultured with same progress as described above. To generate apical-out organoids, organoid fragments were cultured in 500 μL of organoid culture media in ultra-low attachment 24-well cell culture plate (Corning Inc., Corning, NY, USA). For experiments, basal-out and apical-out organoids were cultured for 3 days.

### 2.4. Imaging of Porcine Organoids

For porcine basal-out and apical-out organoid images, inverted or confocal microscope were used in this study. The bright-field images of porcine organoid, such as development of basal organoids and comparison of two types of organoid morphology, were examined under an inverted microscope (Olympus IX51, Tokyo, Japan) daily to check their form (×40 or ×100 magnification). A confocal microscope (K1-Fluo, Nanoscope systems, Daejeon, Korea) was used for all fluorescence images of porcine organoids in the experiments (×400 magnification). The AlexaFlour 555 phalloidin (Thermo Fisher Scientific, Waltham, MA, USA) was used to check for axis change of basal-out and apical-out porcine organoids by F-actin staining. The basal-out organoids were collected by the same method mentioned above, and the apical-out organoids were collected directly. The organoids were fixed with 2% paraformaldehyde for 20 min at room temperature. After washing with PBS two times, samples were stained with 4, 6-Diamidino-2-phenylindole (DAPI) and AlexaFluor 555 phalloidin conjugated to F-actin. DAPI was diluted 1:500 with PBS, and phalloidin was diluted 1:200 with PBS. After staining, the samples were subjected to analysis by confocal microscope.

### 2.5. Quantitative Real-Time PCR (qRT-PCR)

Samples were processed for total RNA extraction using TRIzol™ Reagent (Thermo Fisher Scientific, Waltham, MA, USA). The 500 ng of isolated total RNA was used for cDNA synthesis using AccuPower^®^ RT PreMix (Bioneer, Daejeon, Korea) in accordance with manufacturer’s instructions. qRT-PCR was performed using a QuantStudio 1 Real-Time PCR system (Applied Biosystems, Waltham, CA, USA) with reaction conditions as follows; 50 °C for 10 min, 95 °C for 10 min, 95 °C for 30 s, and 60 °C for 30 s (40 cycles), followed by melting curve analysis. The *GAPDH* gene was used as a housekeeping gene, and relative gene expression level was calculated using the 2^−^^ΔΔC^_t_ method [19]. To confirm intestinal marker gene expression, gel electrophoresis was performed in this study. In brief, the qRT-PCR products were mixed with Dyne LoadingSTAR (Dyne Bio Inc., Seoul, Korea) and were run in gel electrophoresis at 100 V for 30 min by using Agaro-Power™ System (Bioneer, Daejeon, Korea). The primer information used in this study is listed in Table 1.

### 2.6. Assessment for Fatty Acid Uptake

Fatty acid analog absorption assay was performed for two types of organoids. For basal-out organoids assay, Matrigel was solubilized in cell recovery solution at 4 °C for 30 min by using an orbital shaker with gentle, slow shaking (60 rpm). The organoids were washed with advanced DMEM/F12, then fatty acid analog absorption assay was conducted. For absorption test, C1-BODIPY-C12 (Thermo Fisher Scientific, Waltham, MA, USA), which is fluorescent fatty acid analog, was used on porcine intestinal organoids. BODIPY staining of porcine organoids was conducted according to a previously established protocol with a minor modification [16]. In brief, the organoids were resuspended in a solution of 1 μM C1-BODIPY-C12 with 10% fatty-acid-free BSA solution, then incubated in ultra-low attachment 24-well cell culture plate for 30 min. Then, organoids were fixed in 2% paraformaldehyde and stained with DAPI for images. The intracellular fluorescent signal from absorbed C1-BODIPY-C12 was quantified using ImageJ software (NIH).

### 2.7. Breach Epithelial Barrier

Apical-out organoids were harvested and resuspended in advanced DMEM/F12 containing 4 kDa FITC–dextran (2 mg/mL, Sigma-Aldrich, St. Louis, MO, USA). Apical-out organoids were treated with 5 mM EDTA in PBS on ice (for 30 min) and resuspended in the FITC–dextran solution. Then, apical-out organoids were immediately observed under confocal microscope.

### 2.8. Statistical Analysis

All data from experiments were presented as means ± standard deviation (SD). The epithelial cell marker gene expression level was presented using the tissue of each individual pig and organoids formed from them (*n* = 3). The BODIPY uptake of porcine organoids was presented each organoid (*n* = 10) and 7–10 areas were used for calculation. The significance between two groups was analyzed by two-tailed unpaired Student’s *t*-test with Prism 8 software (GraphPad, La Jolla, CA, USA). Statistical significance was considered at *p* < 0.05.

## 3. Results

### 3.1. Development of Porcine Small Intestinal Organoids

Porcine small intestinal organoids were prepared using the scheme of the experimental procedure shown in Figure 1A. The crypts were isolated by EDTA incubation (Figure 1B), and crypts were embedded in Matrigel in organoid culture media containing various factors for organoid development (See Material and Method Section 2.2 for organoid culture media construction). In the organoid culture system, the size of each porcine crypt became larger, and the morphology changed to an intestinal organoid shape via cell proliferation and differentiation. At three to four days after culturing, most of the porcine crypts budded and formed the crypt–villus structure (Figure 1C). These results demonstrate that the porcine jejunum organoid culture system was successfully established as a basal-out model.

### 3.2. Morphological Difference between Porcine Apical-Out and Basal-Out Organoids

Although basal-out organoids were well formed, most of the gut physiological phenomena, such as nutrient absorption, occur in the apical side of gut epithelial cells. Therefore, we used the suspension sub-culture method to expose the apical membranes during the sub-culture protocol. Unlike the basal-out organoid sub-culture method that uses Matrigel, the sub-culture of apical-out organoids is conducted in suspension culture on an ultra-low attachment plate without using Matrigel. Apical-out organoids induced polarity reversal and formed shapes compared with basal-out organoids (Figure 2A). In the sub-cultured basal-out organoids, fragments of organoids were larger and budded over three days of culture and had a clear lumen structure. Unlike basal-out organoids, the apical-out organoids formed a spherical shape and did not form a budding structure. To clarify the structural differences in apical-out and basal-out organoids, we performed staining for each organoid using F-actin (Figure 2B). Strong F-actin staining of microvilli brush borders showed the difference between the outward surface of basal-out and apical-out organoids. To determine whether the two types of porcine organoids have intestinal epithelial cell characteristics, we performed qRT-PCR for examining the mRNA expression for genes of various intestinal epithelial cell types. The major small intestinal epithelial cells include intestinal stem cells, goblet cells, Paneth cells, enterocytes, and enteroendocrine cells. The mRNA expressions of intestinal epithelial cells (*Lgr5* for intestinal stem cells, *Muc2* for goblet cells, *Lyz* for Paneth cells, *ChgA* for enteroendocrine cells, and *ALPI* for enterocytes) were compared with those of porcine jejunum tissue (Figure 3). A comparison between the two types of organoids revealed that *Lgr5* and *Lyz* showed low gene expression, whereas *ChgA* and *ALPI* showed higher gene expression in apical-out organoids than in basal-out organoids. These results imply that both basal-out organoids and apical-out organoids include major intestinal epithelial cell types.

### 3.3. Fatty Acid Analog Uptake by Different Types of Porcine Organoids

One of the major physiological responses that occurs in the small intestine is nutrient absorption. Thus, we determined whether our organoid system was functional for nutrients. We used fatty acid as one of nutrients absorbed in the small intestine. Fatty acids are incorporated into lipid droplets after absorption and transported to the basal regions of the intestinal epithelial cells. To evaluate the fatty acid absorption by porcine organoids, we treated basal-out organoids and apical-out organoids with fluorescent fatty acid analogs (Figure 4A). Strong fluorescent signals and a larger area of lipid droplets were observed in apical-out organoids compared to basal-out organoids (Figure 4B). These data suggest that apical-out porcine organoids may serve as more efficient models for nutrient-related studies.

### 3.4. Porcine Apical-Out Organoids Have Gut Barrier Function

Gut epithelial cells act as a physical barrier for separation between luminal antigen and host immune cells. This barrier function of intestinal epithelial cells is important for maintaining intestinal tissue homeostasis. As shown in Figure 5, strong signals for fluorescent dextran were detected outside the organoids; however, the apical-out organoids completely blocked the inflow of FITC–dextran due to the barrier integrity. Tight junctions and barrier integrity in apical-out organoids can be disrupted by EDTA treatment. FITC–dextran penetrates EDTA-treated apical-out organoids and spreads into the intercellular region of the organoids. These findings confirm that porcine apical-out organoids form an intact epithelial barrier and can be applied as a useful model in gut barrier integrity studies.

## 4. Discussion

The intestinal epithelium acts as a major part of the GIT with multiple physiological functions. For example, gut epithelial cells secrete enzymes to digest food materials and absorb nutrients from the lumen of the intestine. The intestinal epithelium also provides a physical and immunological barrier against luminal antigens including toxins and pathogens [4]. Many studies have used in vitro cell lines to better understand the role of intestinal epithelial cells in the regulation of various intestinal tissue responses including nutrient uptake and gut epithelial barrier function. Multiple types of intestinal epithelial cell lines are available. Caco-2 and IEC-6 cells have been widely used to determine the effects of bioactive compounds on functions of epithelial cells such as gut barrier functions [20]. For example, butyrate increased the TEER of Caco-2 cells and facilitated the assembly of tight junctions through activation of AMP-activated protein kinase [21]. Domestic-animal-originated IPEC-1 and IPEC-J2 have also been used [8,22]. It has been reported that vitamin A regulates tight junction protein (ZO-1, Occludin, Claudin-1) and TEER in IPEC-J2 cells [23]. Jiang et al. used IPEC-J2 cells to determine the effects of 4-Phenylbutyric acid (4-PBA), an aromatic fatty acid, on barrier damage in IPEC-J2 induced by deoxynivalenol (DON) and lipopolysaccharides (LPS) [24]. These gut epithelial cell lines have several advantages in intestinal nutritional and immunological studies. However, there are the limitations of using cell lines in vitro experiment. For example, it is difficult to reproduce a complex tissue system as they are composed of a single type of cell. To make a stable cell line, an immortalization process is required, and this process may affect the normal or intact biological function of this cell line [25]. Thus, it is important to build a better in vitro system that represents complex biological reactions of the intestinal epithelium.

Since the establishment of mouse intestinal organoids in 2009, human and farm animal intestinal organoid culture models have emerged [26,27,28]. The developed organoid culture models have been widely used to investigate GIT mechanisms, such as interactions between pathogens and the host, nutrient absorption, and intestinal epithelium integrity [29,30,31]. Gonzalez et al. demonstrated the first successful intestinal organoid culture in young piglets [12]. Recently, another research team compared expressed genes and pathways in porcine intestinal organoids, epithelial tissues, and IPEC-J2 cells at the transcriptomic level [32]. In brief, their results show that porcine intestinal organoids generally resemble epithelial tissue, not IPEC-J2. Recently, physiological responses of pig organoids have been studied. Glutamate enhances pig organoid development and epithelial cell proliferation via mTORC1 activation, which includes IR, IRS, PI3K, and Akt pathways [33]. Additionally, pig organoids showed a pathophysiological response to cholera toxin, a bacteria-derived toxin [15].

Although an organoid system has various advantages, gut organoids, especially basal-out organoids, have a critical limitation for gut functional studies. In the intestine, the apical side of the intestinal epithelium performs various biological functions such as nutrient absorption and gut microbe recognition. However, the basal-out organoids have limited access to luminal materials such as nutrients and microbes. Thus, researchers often used 2D monolayer organoid culture models to overcome the structural limitations of basal-out organoids [15]. Monolayer pig organoids showed cellular functional transport characteristics, such as SGLT1 and CFTR. In a 2D organoid culture system, an epithelial permeability assay has also been conducted [14]. The tight junction formation within the epithelial monolayer was measured using transepithelial resistance and FITC–dextran. Thus, the 2D monolayer model is useful for intestinal mechanistic studies.

The 2D culture of organoids is an efficient model to study interactive response in the gut tissue, but the in vitro condition still differs from that of the living body. In the biomedical field, human apical-out organoids have been well established for studying barrier integrity, nutrient uptake, and pathogen infection [16,34]. It has been suggested that the apical-out organoid system has certain advantages. For example, apical-out organoids maintain their ability to differentiate into various intestinal epithelial cell types, maintain proper polarity and barrier function, and can absorb nutrients under polarity-specific conditions. The domestic animal apical-out organoid model has also been used to evaluate the pathological response of intestinal epithelial cells to pathogen infection [13]. Li et al. conducted a study on the swine enteric virus and immune responses using a porcine organoid model [35]. They infected the porcine apical-out organoid and investigated the infectivity and antiviral immune responses by identifying interferon response genes, such as *IFN-α*, *IFN-λ1*, *ISG15*, and *IGS58*. In another study, organoids of chickens were studied for host–pathogen interactions, such as bacterial, viral, or protozoal interactions [36]. They established an avian organoid model to investigates the intestines of chickens. Chicken apical-out organoids were used as a model for host–bacterial interactions (*Salmonella typhimurium*), host–viral interactions (influenza A virus), and host–protozoal interactions (*Eimeria tenella*). The apical-out organoid system provides valuable opportunities for studying nutrition and gut barrier function while maintaining the characteristics of the intestinal epithelium and mimicking the intestinal environment.

In this study, we tested the possibility of using apical-out organoids to study nutrient uptake (especially that of fatty acids) and gut barrier integrity, which are major functions of the intestine. Fatty acids are one of the major nutrients absorbed through various transporter proteins located on the apical sides of intestinal epithelial cells [37]. Enterocytes, a major component of intestinal epithelial cells, use specialized receptors and transporters to mediate the uptake of fatty acids at the apical surface [38]. Fatty acids are metabolized within the intestinal epithelial cells after uptake. The triacylglycerols are reformed within the enterocytes from the absorbed fatty acids and are incorporated into chylomicrons. Previous studies have suggested the potential roles of fatty acids in the regulation of intestinal epithelial cells [39]. Fatty acids, including medium-chain and long-chain fatty acids, regulate gut permeability by the activation of nuclear and transmembrane receptors. For example, under inflammatory conditions, *n*-3 polyunsaturated fatty acids (*n*-3 PUFAs) restored impaired gut barrier functions through the regulation of gut permeability, mucus production, and tight junction protein expression [40]. In a pig study, supplementation with butyrate prevented small intestine mucosal atrophy, increased cell proliferation, and decreased apoptosis of enterocytes in neonatal piglets [41]. In this study, we compared the degree of absorption of fatty acid analogs through two porcine organoid models using fluorescence-labeled fatty acids, BODIPY (Figure 4). As expected, apical-out organoids absorbed more BODIPY, forming more lipid droplets compared to basal-out organoids. This observation suggests that the apical-out organoid model is better than the in vitro organoid model system to investigate the biological reactions of fatty acids in the intestinal epithelium. In future studies, other nutrients can be applied to this model based on their physiological reactions in the gastrointestinal tract.

Intestinal epithelial layer lines in the gastrointestinal tract provide a physical barrier (gut barrier function) between host immune cells and microbes in the lumen. Maintaining gut barrier function is critical for regulating gut tissue homeostasis by promoting antibacterial immunity and preventing unnecessary inflammation driven by commensal microbes and food materials. The intestinal epithelial cells regenerate every three to five days; therefore, efficient regeneration of the epithelial layer is necessary to maintain gut barrier function. Gut integrity is determined by a variety of factors, such as intestinal immune cells and the immune system, microbiota, and intestinal epithelium [42]. For example, intestinal epithelial cells regulate gut barrier function by forming tight junctions, thus preventing microorganisms or microbial toxins from penetrating the body [43]. The EDTA solution, which disrupts tight junctions, was used in this study, and it allowed the permeation of FITC–dextran through the intracellular spaces of porcine apical-out organoids (Figure 5). In other words, it indicates a breakdown of the epithelial barrier, and this model can be applied to gut permeability assays. Disruption of epithelial barrier function is associated with many pathological states, including infectious and inflammatory diseases caused by pathogens or toxins. In future studies, the porcine apical-out organoid model can be used to understand the regulatory mechanism of gut permeability and integrity against various environmental factors.

## 5. Conclusions

In this study, we established a porcine basal-out and apical-out organoid system. By performing a comparative study, we proposed a porcine intestinal organoid culture system as a useful model for nutritional or mechanistic studies to elucidate the interrelationship of intestinal epithelial cells. Additionally, we believe that this model has the potential to facilitate future discoveries with respect to farm animals and humans.

## Figures and Tables

**Figure 1 animals-12-00372-f001:**
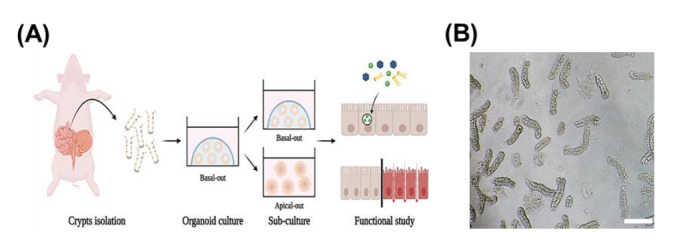
Development of small intestinal organoids in porcine jejunum tissue. (**A**) Brief scheme for experimental procedure of this study. (**B**) Representative image of isolated crypts from pig jejunum tissue (magnification: ×100). Scale bar (100 μm). (**C**) The organoids images show development of organoids from day 0 to day 5 (magnification: ×40). The arrows indicate the organoid budding points. Scale bar (200 μm).

**Figure 2 animals-12-00372-f002:**
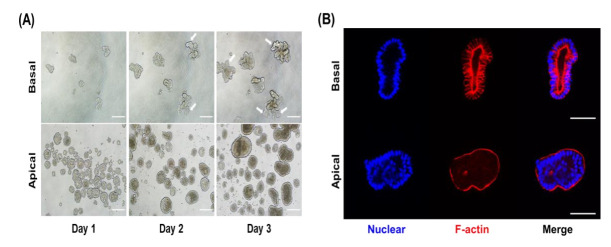
Comparison of morphology for basal-out and apical-out porcine small intestinal organoids. (**A**) Representative images of two types of organoids culture from day 1 to day 3 (magnification: ×100). The arrows indicate the organoid budding points. Scale bar (200 μm). (**B**) Images of stained organoids (magnification: ×400). Nuclear and F-actin were stained by DAPI and phalloidin. The images represent switched axis between basal-out organoid and apical-out organoids. Scale bar (100 μm).

**Figure 3 animals-12-00372-f003:**
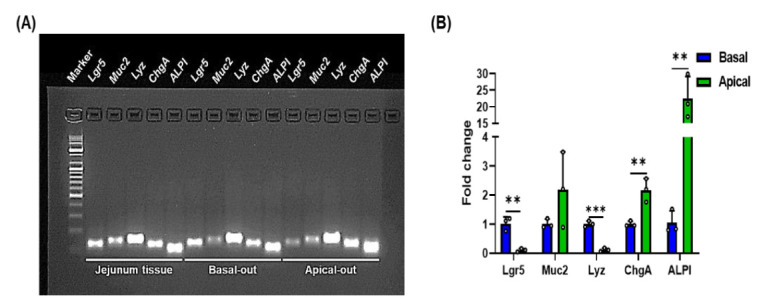
The expression of genes for intestinal epithelial cells in established organoids. (**A**) The image of gel electrophoresis shows expression of intestinal epithelial cell markers (*Lgr5*, *Muc2*, *Lyz*, *ChgA*, *ALPI*). The pig jejunum tissue was used for positive control. (**B**) The mRNA expression of intestinal epithelial cells on both basal-out and apical-out organoids. Data are represented as mean ± SD (*n* = 3). The significance between basal-out and apical-out organoids was analyzed by two-tailed unpaired Student’s *t*-test. ** *p* < 0.01; *** *p* < 0.001.

**Figure 4 animals-12-00372-f004:**
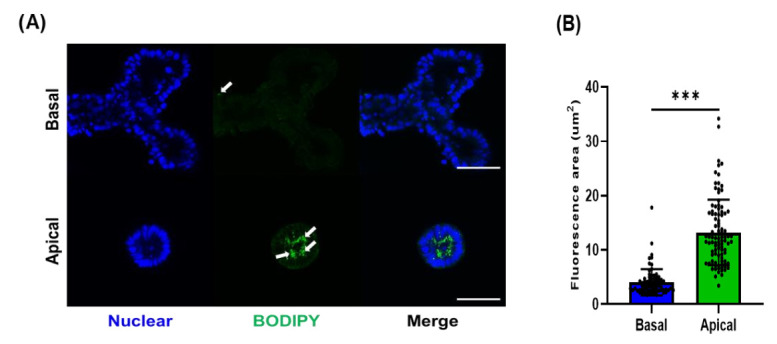
The BODIPY uptake for porcine small intestinal organoid. (**A**) Representative organoid images for basal-out and apical-out organoids treated with fluorescent fatty acid analog (BODIPY) (magnification: ×400). Each organoid was treated with BODIPY for 30 min, then staining was performed. Nuclear was stained by DAPI. Scale bar (100 μm). (**B**) Quantification of fatty acid analog uptake in porcine organoids. Data are represented as mean ± SD (*n* = 10, 7–10 fluorescence area/organoids). The significance between basal-out and apical-out organoid was analyzed by two-tailed unpaired Student’s *t*-test. *** *p* < 0.001.

**Figure 5 animals-12-00372-f005:**
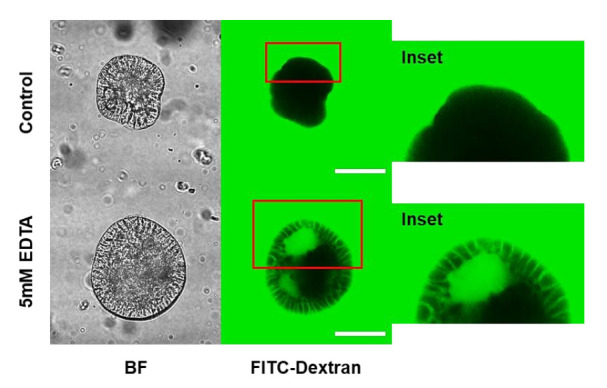
Disrupted gut barrier on porcine small intestinal apical-out organoid. The porcine apical-out organoids were treated in FITC–dextran-containing media (2 mg/mL) (magnification: ×400). Five mM EDTA was used for demonstrating disruption of epithelial barrier integrity. Scale bar (200 μm). BF, bright field.

**Table 1 animals-12-00372-t001:** List of primers used for this study.

Gene	Forward	Reverse	Product Size (bp) ^1^
*Lgr5*	CCTTGGCCCTGAACAAAATA	ATTTCTTTCCCAGGGAGTGG	110
*Muc 2*	GCTGGCCGACAACAAGAAGA	TGGTGGGAGGATGGTTGGAA	126
*Lyz*	GCAAGACACCCAAAGCAGTT	ATGCCACCCATGCTTTAACG	132
*ChgA*	TGAAGTGCATCGTCGAGGTC	GAGGATCCGTTCATCTCCTCG	104
*ALPI*	AGGAACCCAGAGGGACCATTC	CACAGTGGCTGAGGGACTTAGG	83
*GAPDH*	ATTCCACCCACGGCAAGTTC	CACCAGCATCACCCCATTTG	126

^1^ bp; base pair.

## Data Availability

All data presented in this study are available on request from the corresponding author.

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
