# Peer review of "Porcine Intestinal Apical-Out Organoid Model for Gut Function Study"

_animals, 2022, doi:10.3390/ani12030372_

Round 1

Reviewer 1 Report

The paper was well-written, the research was well-designed and discussed. However, some corrections need to be done in order it can published, as follow:

Introduction

Line 58: Please, change "in vivo" to italics. The same in lines 64, 80, 323, 327, 328, 336, 353, 377, 381, 387, 413. Check it out again through the text.

Line 61: Please, change "owing to..." to "due to"

Material and Methods

Line 106: Please correct "fregments" to "fragments"

Line 109: Please correct " "50 ml tube" to "50-mL tube"

Line 110: Please correct "200 g" to "200 x g". The same in line 147.

Line 130: Please change "were used" to "were utilized"

Line 141: Please correct the phrase "Briefly, cells were grown for 4 days (approximately to 8-90% confluency) and take off media (first batch). Then, replace fresh culture media and culture for another 3 days and take off media (second batch)." to "Briefly, cells were grown for 4 days (approximately to 8-90% confluency) and taken off media (first batch). Then, replaced by fresh culture media and cultured for another 3 days and taken off media (second batch)."

Line: 146: Please specify the conditions for "gentle horizontal shaking" in the text. The same in line 188.

Line 147: Please correct "supernatant with organoids were collected and..." to "supernatant with organoids was collected and..."

Lines 161/162: Please specify the fluorescent dye used in text.

Line 180: Please change "...and were electrophoresis at 100 V for 30 min..." to "...and were run in gel electrophoresis at 100 V for 30 min..."

Statistical analysis

Line 205: Please, correct "Student's t test" to "Student's t-test" in text. The same in lines 269, 288.

Results

Line 213: Please specify the "...various factors for organoid devel-opment..." providing the manufacturer information or if it was formulated by researchers, provide it by means of an annex at the end of article or between parenthesis in the text. 

Discussion

Line 387: Please provide the references that allowed the statement to be substantiated by authors. The same in line 395 regarding livestock studies.

Lines 439/440: Please alter the phrase to "Thus, the apical-out organoid model of the present study can be effectively used to investigate the absorption and mechanism of the reaction of nutrients within the intestinal epithelium."

Conclusions

Line 461: Please alter the phrase to the suggested one or reformulate the conclusions section in order to be as clear and succinct as possible: "A porcine intestinal organoid culture system was developed, optimized and provided various research possibilities, such as for future nutritional or mechanistic studies to elucidate the  interrelationship with intestinal epithelial cells. Porcine apical-out organoids can be used as relevant and accessible models for understanding intestinal physiological research. Also, authors believe that this model has the potential to facilitate future discoveries with respect to farm animals and biomedical fields."

Author Response

Reviewer 1

Comments and Suggestions for Authors

The paper was well-written, the research was well-designed and discussed. However, some corrections need to be done in order it can published, as follow:

Introduction

Line 58: Please, change "in vivo" to italics. The same in lines 64, 80, 323, 327, 328, 336, 353, 377, 381, 387, 413. Check it out again through the text.

Line 61: Please, change "owing to..." to "due to"

Material and Methods

Line 106: Please correct "fregments" to "fragments"

Line 109: Please correct " "50 ml tube" to "50-mL tube"

Line 110: Please correct "200 g" to "200 x g". The same in line 147.

Line 130: Please change "were used" to "were utilized"

Line 141: Please correct the phrase "Briefly, cells were grown for 4 days (approximately to 8-90% confluency) and take off media (first batch). Then, replace fresh culture media and culture for another 3 days and take off media (second batch)." to "Briefly, cells were grown for 4 days (approximately to 8-90% confluency) and taken off media (first batch). Then, replaced by fresh culture media and cultured for another 3 days and taken off media (second batch)."

Line 147: Please correct "supernatant with organoids were collected and..." to "supernatant with organoids was collected and..."

Line 180: Please change "...and were electrophoresis at 100 V for 30 min..." to "...and were run in gel electrophoresis at 100 V for 30 min..."

Statistical analysis

Line 205: Please, correct "Student's t test" to "Student's t-test" in text. The same in lines 269, 288.

Response: Thanks for kind corrections. As followed by your suggestions, we have corrected the sentences or words. Please, find all corrections in the revised manuscript

Italic issue

Line: 57, 63, 67, 71, 77, 93, 312, 325, 328, 350, 355, 393 (italic)

Material and Methods

Line: 104, 106, 107, 128, 138-141, 146-148, 147, 180-181, 209, 271, 290

Line: 146: Please specify the conditions for "gentle horizontal shaking" in the text. The same in line 188.

Lines 161/162: Please specify the fluorescent dye used in text.

Results

Line 213: Please specify the "...various factors for organoid devel-opment..." providing the manufacturer information or if it was formulated by researchers, provide it by means of an annex at the end of article or between parenthesis in the text. 

Response: Thanks for comments. We have provided necessary for information (shaking condition, fluorescent dye, and organoid culture condition) in the revised manuscript. Please find all changes in the revised manuscript.

Material and Methods

Line: 145-146, 161-163, 188

Results

Line: 217-218

Discussion

Line 387: Please provide the references that allowed the statement to be substantiated by authors. The same in line 395 regarding livestock studies.

Response: Thanks for suggestion. We added two references to provide the information for apical-out organoid (Line 357, 362).

[34] Han, X.; Mslati, M.; Davies, E.; Chen, Y.; Allaire, J.M.; Vallance, B.A. Creating a more perfect union: Modeling intestinal bacteria-epithelial interactions using organoids. CMGH Cell. Mol. Gastroenterol. Hepatol.2021, 12, 769-782.

[13] Beaumont, M.; Blanc, F.; Cherbuy, C.; Egidy, G.; Giuffra, E.; Lacroix-Lamandé, S.; Wiedemann, A. Intestinal organoids in farm animals. Vet. res. 2021, 52, 1-15.

Lines 439/440: Please alter the phrase to "Thus, the apical-out organoid model of the present study can be effectively used to investigate the absorption and mechanism of the reaction of nutrients within the intestinal epithelium."

Conclusions

Line 461: Please alter the phrase to the suggested one or reformulate the conclusions section in order to be as clear and succinct as possible: "A porcine intestinal organoid culture system was developed, optimized and provided various research possibilities, such as for future nutritional or mechanistic studies to elucidate the interrelationship with intestinal epithelial cells. Porcine apical-out organoids can be used as relevant and accessible models for understanding intestinal physiological research. Also, authors believe that this model has the potential to facilitate future discoveries with respect to farm animals and biomedical fields."

Response: Thank you for your comment. To improve clarity of sentence, we have revised this content. Please, find changes in the revised manuscript.

Line 392-394: Discussion part

Line 416-420: Conclusion part

Reviewer 2 Report

General:

The manuscript submitted by Joo et al. focuses on a relevant field of research on gastrointestinal functions in pigs. Organoids of livestock, especially of pigs are of great interest to generate in vitro models mimicking physiological characteristics of the in vivo situation. Despite the good study design several major comments must be considered to improve the quality of the manuscript.

Title:

Since only the uptake of fatty acids are monitored the phrase “Model for Nutrition” is not adequte and must be avoided.

Introduction:

Line 52 onwards: the mentioned cell types are not expressed in all segments of the GIT, a specification is needed.

Line 64: Costs of an experiment should never be an issue to perform experiments, the suitability of the chosen model should always be the major point of focus. Moreover, in vivo experiments in pig are not necessarily more expensive than alternative methods

Line 76: in the publication of Hoffmann et al. (2021), porcine organoids of the jejunum were used to investigate the glucose absorption and chloride secretion, cellular composition and as proof-of-principle an incubation with cholera toxin to access tissue reaction to bacterial enterotoxins. This publication as well as other publications such as Beaumont et al. (2021) should be included and discussed regarding the own results.

Material and Methods:

Line 101: Since 5 weeks old pigs are undergoing a critical phase of GIT development often characterised by diarrhea after weaning these young pigs are not optimal to generate organoids to mimic a fully developed GIT. In addition, at this age major functional intestinal properties are still under development.

Line 115: what was the total volume of Matrigel drops seeded in the 96 well plates?

Line 127: Typo in SB202190

Line 137: Please use adequate references for the generation of the conditioned media.

Line 150: Which volume of medium was used to cultivate apical-out organoids in 24-well plates?

Lines 159 and 160: what was the numeric aperture of the optical systems used?

Lines 170-183: If GAPDH was used as a housekeeper gene, please mention how the individual genes were normalized to GAPDH.

Lines 203-207: Please mention how biological and technical replicates were generated.

Results:

Line 233 onwards: Figures of basal-out organoids (Fig. 2A) show a poor visibility compared to the apical-out organoids of the same figure. Since figure 1C shows the good visibility of basal-out organoids, comparable pictures should be used in figure 2. F-Actin staining of apical-out organoids is only visible on the far left side of the organoid but not at the complete surface area. Figures that are more representative should be used. Moreover, cellular composition of both organoid types can be investigated using IF and not only gene expression levels.

Line 237: Only by bright field microscopy it is hard to tell that apical-out organoids do not develop a lumen.

Line 248 onwards: Both types of organoids show genes which are also expressed in the native jejunum. However, a comparison to the native jejunum is also performed in a qualitative but not quantitative manner and should be included in Figure 3B. This would further show the comparability between organoids and native tissue.

Line 280-288: The authors only showed the ability of the apical-out organoids to transport fatty acids, not nutrients in general. Thus, most relevant physiological properties have not been studied. Moreover, IF images of BODIPY show very high background, more specific figures should be used.

Discussion:

In general: The authors should restructure and shorten the discussion since several fields of research are discussed intensively which are not the scope of this study. Moreover, results of own data must be discussed in more depth.

Lines 308-337: This passage must be shortened and restructured. Since beneficial bacteria is not the scope of this manuscript, one should refer to nutrient uptake studies of Caco-2 and HT29-MTX cells.

Line 379: Remove the sentence “However, this method cannot be applied to conventional laboratories.”

Lines 452-459: Since EDTA is often used as positive control to disrupt cellular junctions, it may be the case that this model can be used for gut permeability assays but needs further validation.

Beaumont, M., Blanc, F., Cherbuy, C. et al. (2021). Intestinal organoids in farm animals. Veterinary Research 52, 33. doi:10.1186/s13567-021-00909-x

Hoffmann, P., Schnepel, N., Langeheine, M. et al. (2021). Intestinal organoid-based 2d monolayers mimic physiological and pathophysiological properties of the pig intestine. PLOS ONE 16, e0256143. doi:10.1371/journal.pone.0256143

Author Response

Reviewer 2

Comments and Suggestions for Authors

General:

The manuscript submitted by Joo et al. focuses on a relevant field of research on gastrointestinal functions in pigs. Organoids of livestock, especially of pigs are of great interest to generate in vitro models mimicking physiological characteristics of the in vivo situation. Despite the good study design several major comments must be considered to improve the quality of the manuscript.

Title:

Since only the uptake of fatty acids are monitored the phrase “Model for Nutrition” is not adequte and must be avoided.

Response: Thank you for suggestion. We agreed your opinion that our study contains limited information for nutrition part. So, we modified manuscript title to “Porcine Intestinal Apical-out Organoid Model for Gut Function Study”. However, if you have better idea, we like to hear. Thanks again.

Introduction:

Line 52 onwards: the mentioned cell types are not expressed in all segments of the GIT, a specification is needed.

Response: We appreciated your comment. We have corrected the phrase to “For example, ISCs in the small intestine are able to differentiate into specific cell types, such as absorptive enterocytes, goblet cells, Paneth cells, and enteroendocrine cells.” (Line 51-53).

Line 64: Costs of an experiment should never be an issue to perform experiments, the suitability of the chosen model should always be the major point of focus. Moreover, in vivo experiments in pig are not necessarily more expensive than alternative methods

Response: Thank you for raising important issue. We agreed with your opinion that cost is not important factor for designing experiment. We have corrected this sentence to provide information on limitation of in vivo study like difficulties in manipulating experiment, high-throughput study, and animal welfare issue. Please, find changes in revised manuscript (Line 63-65).

Line 76: in the publication of Hoffmann et al. (2021), porcine organoids of the jejunum were used to investigate the glucose absorption and chloride secretion, cellular composition and as proof-of-principle an incubation with cholera toxin to access tissue reaction to bacterial enterotoxins. This publication as well as other publications such as Beaumont et al. (2021) should be included and discussed regarding the own results.

Response: Thank you for providing valuable references. We have added findings from Hoffmann et al., (2021) and Beaumont et al., (2021) in the introduction part. And we have added more contents with these references in discussion part as well. (Line 76, 84, 342, 349, 362)

Material and Methods:

Line 101: Since 5 weeks old pigs are undergoing a critical phase of GIT development often characterised by diarrhea after weaning these young pigs are not optimal to generate organoids to mimic a fully developed GIT. In addition, at this age major functional intestinal properties are still under development.

Response: Thank you for your comments. In this study, we used healthy piglets without diarrhea symptoms. We have added information about body weight, and diarrhea symptoms for piglets used in study. Also, we understand your point that the intestine of 5 weeks old pigs is under developing. However, several porcine organoid studies used relatively young pigs. For example, Khalil et al. (2016) used 4- to 8-week-old mini-Yucatan pigs and van der hee et al. (2020) used 5-week-old piglets for organoid research. We used 5-weeks-old piglets in this study referring to these references as followed.

[12] Khalil, H.A.; Lei, N.Y.; Brinkley, G.; Scott, A.; Wang, J.; Kar, U.K.; Jabaji, Z.B.; Lewis, M.; Martín, M.G.; Dunn, J.C.Y. et al. A novel culture system for adult porcine intestinal crypts. Cell Tissue Res. 2016, 365, 123-134.

[32] van der Hee, B.; Madsen, O.; Vervoort, J.; Smidt, H.; Wells, J.M. Congruence of transcription programs in adult stem cell-derived jejunum organoids and original tissue during long-term culture. Front. Cell Dev. Biol. 2020, 8, 375.

Line 115: what was the total volume of Matrigel drops seeded in the 96 well plates?

Response: We used 4 μL of Matrigel for each well in the 96-well plates. We added information. (Line 114)

Line 127: Typo in SB202190

Response: Sorry for minor error. We corrected this. (Line 123)

Line 137: Please use adequate references for the generation of the conditioned media.

Response: We added reference for the R-spondin 1 and Wnt-3A cell conditioned media.

[17] Li, X.-G.; Zhu, M.; Chen, M.-X.; Fan, H.-B.; Fu, H.-L.; Zhou, J.-Y.; Zhai, Z.-Y.; Gao, C.-Q.; Yan, H.-C.; Wang, X.-Q. Acute exposure to deoxynivalenol inhibits porcine enteroid activity via suppression of the Wnt/β-catenin pathway. Toxicol. Lett. 2019, 305, 19-31.

Line 150: Which volume of medium was used to cultivate apical-out organoids in 24-well plates?

Response: We used 500 μL of organoid culture media for apical-out organoid culture in 24 well-plates. We have added information for volume of media. (Line 151)

Lines 159 and 160: what was the numeric aperture of the optical systems used?

Response: We used x 40 magnification for Figure 1C and x 100 magnification for Figure 1B and Figure 2A. All fluorescence organoid images in Figure 2B, Figure 4A, Figure 5 were obtained by using x 400 magnification. We added this information in M&M and legends for figures (Line 159, 161, 228, 229, 259, 261, 286, 304).

Lines 170-183: If GAPDH was used as a housekeeper gene, please mention how the individual genes were normalized to GAPDH.

Response: We used GAPDH gene as a housekeeping gene and normalized it by using the 2-ΔΔCt method. We added calculation method for relative gene expression level calculation with suitable reference in M&M section (Line 177-178).

[19] Schmittgen, T.D.; Livak, K.J. Analyzing real-time PCR data by the comparative CT method. Nat. Protoc. 2008, 3, 1101-1108.

Lines 203-207: Please mention how biological and technical replicates were generated.

Response: We added replication information to M&M section for qRT-PCR and BODIPY uptake experiments conducted in this study (Line 206-208).

Results:

Line 233 onwards: Figures of basal-out organoids (Fig. 2A) show a poor visibility compared to the apical-out organoids of the same figure. Since figure 1C shows the good visibility of basal-out organoids, comparable pictures should be used in figure 2. F-Actin staining of apical-out organoids is only visible on the far left side of the organoid but not at the complete surface area. Figures that are more representative should be used. Moreover, cellular composition of both organoid types can be investigated using IF and not only gene expression levels.

Response: Thank you for comment. We have added new image for Figure 2A and 2B with better visibility. We agreed your opinion that we can stain for organoids with other epithelial cell biomarkers such as Lgr5, Muc2, Lyz, ChgA, ALPI. Unfortunately, we could not show more IF staining image with these biomarkers because we do not have antibodies for these at this stage. If you think, it is really necessary, we will try it in next round of revision. Please, understand our limitation in the current stage of revision.

Line 237: Only by bright field microscopy it is hard to tell that apical-out organoids do not develop a lumen.

Response: Thank you for your comment. We revised contents to prevent confusion on apical-out organoid structure. (Line 238-241)

Line 248 onwards: Both types of organoids show genes which are also expressed in the native jejunum. However, a comparison to the native jejunum is also performed in a qualitative but not quantitative manner and should be included in Figure 3B. This would further show the comparability between organoids and native tissue.

Response: Thank you for valuable comment. As you suggested, we have examined the epithelial cell marker gene expression in pig jejunum tissue to compare. As you can see the figure below, they express all biomarker for epithelial cells, they seem to have very different levels of expression. In general, tissue has much higher expression in MUC2 and Lysozyme. One of possible explanation for this discrepancy is that they have very different cellular composition. For example, intestinal tissue has laminar propria area, where various immune cells densely located and they known to affect the antimicrobial biomaterials via type I interferon signaling (goblet and Paneth cells) (Tschurtschenthaler et al., 2014). Although, gut organoid system is good model to mimic the gut tissue, we assume that gut organoids do not really represent whole gut tissue. If you think tissue data is good to add in the main figures, we would replace it.

Tschurtschenthaler, M.; Wang, J.; Fricke, C.; Fritz, T.M.; Niederreiter, L.; Adolph, T.E.; Sarcevic, E.; Künzel, S.; Offner, F.A.; Kalinke, U. et al. Type I interferon signalling in the intestinal epithelium affects Paneth cells, microbial ecology and epithelial regeneration. Gut 2014, 63, 1921-1931.

Line 280-288: The authors only showed the ability of the apical-out organoids to transport fatty acids, not nutrients in general. Thus, most relevant physiological properties have not been studied. Moreover, IF images of BODIPY show very high background, more specific figures should be used.

Response: Thanks for great comment. We used C1-BODIPY-C12, which is a fluorescence fatty acid analog. It is a test compound and has been used for lipid trafficking studies. In our study, we tried to show that porcine apical-out organoid model can be used more efficiently in nutrition research such as fatty acid absorption than conventional organoid model. In relevant to this tissue, we revised title of this manuscript and added more information for limitation of our study in the discussion section (Line 394-396). Please, find the revised manuscript. Also, IF images of Figure 4A were changed by that show more specific and representative one.

Discussion:

In general: The authors should restructure and shorten the discussion since several fields of research are discussed intensively which are not the scope of this study. Moreover, results of own data must be discussed in more depth.

Response: Thank you for suggestion. As we understand your points, we have reduced unnecessary contents and revised it to deal with core of findings in this study. Please, find the revised manuscript.

Lines 308-337: This passage must be shortened and restructured. Since beneficial bacteria is not the scope of this manuscript, one should refer to nutrient uptake studies of Caco-2 and HT29-MTX cells.

Response: Thanks for suggestion. We reconstructed the paragraph that you mentioned. Now, they have contents with importance of intestine, limitations and necessity of cell line-based research and organoid system. (Line 324-329)

Line 379: Remove the sentence “However, this method cannot be applied to conventional laboratories.”

Response: We remove the phrase in the revised manuscript.

Lines 452-459: Since EDTA is often used as positive control to disrupt cellular junctions, it may be the case that this model can be used for gut permeability assays but needs further validation.

Response: Thank you for comment. Pig apical-out organoid model can be used not only permeability test, but also gut barrier disruption factors (like pathogens and toxins). Discussion section was modified to emphasize the potential research possibility for pig apical-out organoid model (Line 412-414).

Beaumont, M., Blanc, F., Cherbuy, C. et al. (2021). Intestinal organoids in farm animals. Veterinary Research 52, 33. doi:10.1186/s13567-021-00909-x

Hoffmann, P., Schnepel, N., Langeheine, M. et al. (2021). Intestinal organoid-based 2d monolayers mimic physiological and pathophysiological properties of the pig intestine. PLOS ONE 16, e0256143. doi:10.1371/journal.pone.0256143

Reviewer 3 Report

Dear Authors,

I wish to commend you for the well prepared manuscript and the efforts made to make it concise and relevant to the audience of this journal. There are a few minor comments in the attached for your consideration.

Author Response

Reviewer 3

Comments and Suggestions for Authors

Dear Authors,

I wish to commend you for the well prepared manuscript and the efforts made to make it concise and relevant to the audience of this journal. There are a few minor comments in the attached for your consideration.

Response: Thank you for your kind comments. Following your opinions, we have corrected sentences and added references to revised discussion part. We also have modified the conclusions section to clearly convey implication of this study.

Response to report

Line 148: We corrected “with a pipetting” to “with pipetting”.

Line 319(non-revised manuscript): We removed the reference while revising the manuscript.

Line 314-319: Appropriated references were added to the revised manuscript.

[8] Ziegler, A.; Gonzalez, L.; Blikslager, A. Large animal models: the key to translational discovery in digestive disease research. CMGH Cell. Mol. Gastroenterol. Hepatol. 2016, 2, 716-724.

[20] Couto, M.R.; Gonçalves, P.; Catarino, T.A.; Martel, F. The effect of inflammatory status on butyrate and folate uptake by tumoral (Caco-2) and non-tumoral (IEC-6) intestinal epithelial cells. Cell J 2017, 19, 96.

[22] Diesing, A.-K.; Nossol, C.; Panther, P.; Walk, N.; Post, A.; Kluess, J.; Kreutzmann, P.; Dänicke, S.; Rothkötter, H.-J.; Kahlert, S. Mycotoxin deoxynivalenol (DON) mediates biphasic cellular response in intestinal porcine epithelial cell lines IPEC-1 and IPEC-J2. Toxicol. Lett. 2011, 200, 8-18.

Line 16-420: Thank you for your comment. To improve clarity of sentence, we have revised conclusion section. Please, find changes in the revised manuscript.

Reviewer 4 Report

Need to include N in methods, starting with number of pigs.

According to methods, used C1-BODIPY-C12, which is a labeled fatty acid analog of molecular wt 412. This is a test compound that suggests, but does not confirm, uptake of the wide range of fatty acids. Results and Conclusion should be modified to reflect this.

Author Response

Reviewer 4

Comments and Suggestions for Authors

Need to include N in methods, starting with number of pigs.

Response: Thank you for comments. In this study, total 3 piglets were used and they were healthy status without diarrhea symptoms. In other to convey more details, phrase was added about animal number, body weight, and diarrhea symptoms used in study. (Line 97-98)

According to methods, used C1-BODIPY-C12, which is a labeled fatty acid analog of molecular wt 412. This is a test compound that suggests, but does not confirm, uptake of the wide range of fatty acids. Results and Conclusion should be modified to reflect this.

Response: Thank you for your valuable comment. In our study, we tried to show that porcine apical-out organoid model can be used more efficiently in nutrient research such as fatty acid absorption than conventional organoid model (Basal-out organoids) through visualization using BODIPY. The limitation of our study you mentioned were added in the revised manuscript (Line 412-414).

Response to report

Line 114: We added concentration information using parenthesis.

Line 109: We already wrote full name of “advanced DMEM/F12” to “advanced Dulbecco’s modified Eagle medium/F12”.

Line 131: We corrected citation of reference [18].

Line 191: We revised “fatty acid” to “fatty acid analog”.

Line 284: We corrected “fatty acid uptake” to “BODIPY uptake” and also “fluorescent fatty acid” to “fluorescent fatty acid analog” in Figure legend.

Round 2

Reviewer 2 Report

The comments have been considered appropriately.